

# The influences of incorporating dynamical external forcing in WRF v3.8.1 on regional climate simulation in China

Jinming Feng[1*], Meng Luo[2,3*], Jun Wang[1], Yuan Qiu[1], Qizhong Wu[4], Ke Wang[5]

[1]Key Laboratory of Regional Climate-Environment for Temperate East Asia, Institute of Atmospheric Physics, Chinese Academy of Sciences, Beijing 100029, China

[2]Yunnan Climate Center, Kunming 650034, China

[3]Yunnan Key Laboratory of Meteorological Disasters and Climate Resources in the Greater Mekong Subregion, Yunnan Provincial Climate Center, Kunming, 650000, China

[4]College of Global Change and Earth System Science, Beijing Normal University, Beijing 100875, China

[5]Tianjin Central Observatory for Oceanic Meteorology, Port and Shipping Meteorological Service Centre of Tianjin, Tianjin 300074, China

*Correspondence to*: Jinming Feng (fengjm@tea.ac.cn), Meng Luo (mrluolang@126.com)

**Abstract:** External forcing is the driving force of climate system, which has significant impact on long-term climate changes. Unlike in the Global Climate Models whose external forcing is clearly prescribed, whether or not to use spatial-temporal varying external forcing to force the Regional Climate Models (RCMs) is still lacking evaluations. Here we modify the regional climate model

WRF v3.8.1 to include all kinds of spatial-temporal varying external forcing components, and further investigate the impact of dynamical forcing on the long-term simulation in China. The results showed that different external forcing configurations in WRF could result in a variation range of 0.08 °C/10a for annual temperature trend and 19.9 mm/10a for annual precipitation trend in Eastern China (EC), whose impact was stronger than parameterization schemes but was weaker

than spectral nudging. The influence of spectral nudging on long-term trend also depended on the configuration of parameterization schemes and external forcing. The WRF model could reasonably reproduce the forcing response pattern of temperature, precipitation, and associated radiative and circulation anomaly changes. The forced annual temperature trend in China could be roughly explained by the linear superposition of GHGs and anthropogenic aerosols, while the

forcing response pattern of summer precipitation trend was mainly determined by anthropogenic aerosols. Therefore, we recommend that when using RCMs for long-term simulations, one should first run a long-term preliminary test to determine whether or not to use nudging, and it is better to use all set of varying forcing components than to use varying GHGs only.

**Key words:** Regional Climate Models (RCMs); WRF; External forcing; Greenhouse Gases (GHGs); Anthropogenic aerosols; Forcing response

## 1 Introduction

External forcing refer to the climate change driving factors outside the climate system, and
could be divided into anthropogenic forcings such as greenhouse gases (GHGs, eg., Manabe and Wetherald, 1980), anthropogenic aerosols (Bollasina et al., 2011), tropospheric Ozone (Shindell et



al., 2006), land use and land cover change (Kalnay and Cai, 2003), etc., as well as natural forcings such as solar activities (Gray et al., 2010) and volcanic aerosols (Robock, 2000). Numerous studies have revealed that external forcing can significantly influence the climate variability on

various temporal scales ranging from seasonal/interannual to decadal/multi-decadal and even longer time scales, especially for the long-term climate change (e.g., Burke and Stott, 2017; Chen et al., 2020a; Hegerl et al., 2011; Li et al., 2021; Meehl et al., 2004; Wang et al., 2013; Wilcox et al., 2013). What's more, some forcing components (such as anthropogenic aerosols, land use and land cover change) have large spatial heterogeneity (Leibensperger et al., 2012; Liu and Matsui,

2021; Zhang et al., 1996), the climate responses to forcing may be different among regions (Kelley et al., 2012; Kirchmeier-Young and Zhang, 2020; Song et al., 2014), and the sensitivity to forcing may also strongly depends on regional features (Coopman et al., 2018; Najafi et al., 2015; Paeth et al., 2009). So far, it is still challenging to evaluate the impact of external forcing on regional climate change.

Regional Climate Models (RCMs) are powerful tools to research climate change at regional scale. Comparing with General Circulation Models (GCMs), RCMs usually have higher resolutions, and can present more detailed information for local and regional scales, which may compensate for the shortcomings of global models (Feser et al., 2011; Giorgi and Gutowski, 2015; Liang et al., 2012; Prommel et al., 2009; Xu et al., 2019). RCMs have been widely used in

sensitivity tests (Pavlidis et al., 2020; Takahashi et al., 2010; Wang et al., 2018; Zou et al., 2014), historical simulations (Gao, 2020; Mariotti and Dell'Aquila, 2011; Yu et al., 2015), future projections (Gao et al., 2013; Martel et al., 2020; Zhu et al., 2018), and even paleoclimate simulations (Fallah et al., 2018; Gomez-Navarre et al., 2013; Strandberg et al., 2022), etc. However, there are many factors that may have significant impact on the RCMs simulations,

which may increase the uncertainties when explaining the results. For instance, the changes in lateral boundary conditions (Diaconescu and Laprise, 2013; Xue et al., 2014), domain size (Karmacharya et al., 2015; Seth and Giorgi, 1998), spatial resolution (Maurya et al., 2018; Tramblay et al., 2013), parameterization schemes (Hui et al, 2019; Niu et al., 2020; Sun and Liang, 2020), and nudging (Spero et al., 2014; Tang et al., 2017; Von Storch et al., 2000) may resulted in

quite different simulation results for the same RCM.

Although the above-mentioned settings in RCMs have been tested in many works, the evaluations for the effect of using external forcing inside the RCMs are still rarely seen. For instance, the COordinated Regional Downscaling EXperiment (CORDEX, Gutowski et al., 2016) supported by the World Climate Research Program (WCRP) has recommended the participators to

use the same forcing configurations between the lateral boundary conditions and inside the RCMs, but not presented related literatures and supporting information. Some researchers argue that using identical forcing configurations inside RCMs and in lateral boundary conditions can avoid physical inconsistencies (e.g., Gomez-Navarro et al., 2013), but still no related evaluations. Jerez et al. (2018) investigated the effect of using GHGs forcing inside RCMs on the surface air

temperature trend in Europe both in the historical and future projection simulations. They found that the GHGs forcing inside the RCMs can resulted in a 1~2 K temperature trend increase on the centennial scale, and in some region the warming signal has been doubled. It implies that the impact of GHGs forcing in the RCMs on temperature trend cannot be neglected in Europe. They also suggest that one should pay cautious when using GHGs forcing in the RCMs, at least the

forcing configurations should be clearly documented for comparisons and repetitions. Boé et al.



(2020) compared the summer climate simulation results in Europe from RCMs and GCMs. Their results showed that the solar radiation change differences between RCMs and GCMs were mainly came from the absence of time-varying anthropogenic aerosols in most RCMs. However, these studies did not consider all of the forcing components, and the results may also depend on the region selections.

China is one of the most densely populated areas in the world, and its monsoon climate are strongly modulated by internal climate variabilities (Huang et al., 2013; Luo et al., 2020; Xue et al., 2015; Zhou et al., 2009; Zhu et al., 2016). In addition, the human activities in China are also very active, which resulted in large amount of GHGs and anthropogenic aerosol emissions as well as obvious land use and land cover changes (Qin and Xie et al., 2012; Song and Deng, 2017; Wang and Zhao 2015; Zhang et al., 2012). Many studies have employed RCMs to research the climate change in China (Feng et al., 2015; Gao et al., 2012; Li et al, 2018; Wang and Kong, 2021; Wang et al., 2019; Yan et al., 2019). However, the knowledge about the impacts of using spatial-temporal varying external forcing in RCMs on the climate simulation in China are still very limited at present.

To deal with such situations, we use a modified Weather Research and Forecasting Model (WRF) version 3.8.1 to include all kinds of varying external forcing, and choose China as our main research region, particularly focus on the Eastern China (EC, 105-123°E, 20-45°N) which has most populations and is strongly affected by monsoon climate, with the purpose of resolving the following questions: 1) to what extent does the forcing configuration in WRF influence the long-term trend simulation results in China, and the comparison with other influencing factors such as parameterization schemes and nudging; 2) the individual impact of some single forcing components (GHGs, anthropogenic aerosols, volcanic aerosols) on simulation results in China; and 3) whether the forcing responses of WRF model are reasonable.

## 2 Model and Data

### 2.1 Model improvements

The present version of WRF has only considered dynamical GHGs forcing, while other forcing components such as anthropogenic aerosols, ozone, and solar constant are usually presented in the form of prescribed climatological mean values in some physical parameterization schemes. In addition, the effect of volcanic aerosols are not considered in all of the physical schemes in WRF v3.8.1. To introduce the spatial-temporal varying external forcing in the WRF model, we have made some modifications in WRF. We revised the Thompson microphysics scheme and the RRTMG radiation transfer scheme to bring in the spatial-temporal varying aerosol concentrations. Aerosols have been divided into two types to fit the Thompson microphysics scheme, i.e., the water-friendly aerosols (consist of sulfate aerosols, sea salt, and organics aerosols) and the ice-friendly aerosols (mainly dust aerosols). In our study region, water-friendly aerosols can generally be treated as anthropogenic aerosols, and ice-friendly aerosols can be treated as natural source aerosols. The varying volcanic aerosols, ozone, and solar constant are added in the RRTMG radiation physical scheme. The Noah/Noah-MP land surface model has been revised to take into account the changing land use and land cover according to the dynamical vegetation settings in the CLM4 land surface model. All these external forcing components are prescribed and made into the auxiliary input files for the ARW solver of WRF model.



**2.2 Data**

The gridded observation data used for model evaluation were the CN05.1 (Wu and Gao, 2013), and we also use another dataset, the CRU TS4.01 (Harris et al., 2014), for cross validation. These two datasets were widely used in climate research and model evaluation studies. The lateral boundary conditions used for driving WRF were the ERA40 reanalysis data (Uppala et al., 2005). 135 The external forcing data were identical to the CMIP5 historical experiment (Taylor et al., 2012), including global mean GHGs series ($CO_2$, $CH_4$, $N_2O$) and solar constant series, as well as spatial-temporal varying aerosols (sulfate, sea salt, organics, dust), ozone, and volcanic aerosols (Figure S1, Figure S2). The land use/land cover change data was originated from the dynamical vegetation data of the CLM4 land surface model, and it was transformed to the classification of 140 IGBP-MODIS in WRF (Figure S3, see Lawrence et al., 2011).

**2.3 Experiment design**

We designed a set of sensitive experiment to investigate various factors affecting the long-term simulation using RCM (**Table 1**). The ERA40 reanalysis data was used as initial and lateral 145 boundary condition for all the experiments. The domain in this study was 0–55°N, 57.5–142.5°E, and the ten grids near the boundary were treated as the buffer zone. The horizontal and vertical resolution were 0.5°×0.5° and 35 levels (top level at 20hPa), respectively. The simulation time ranging from January 1958 to August 2002, and the first year was treated as the spin-up period. The common parameterization schemes used in each experiment were the RRTMG radiation scheme 150 (Iacono et al., 2008), the Thompson aerosol-aware microphysics scheme (Thompson and Eidhammer, 2014), and the Noah_MP land surface model. Two assemble of physical parameterization schemes using different Planetary Boundary Layer (PBL) scheme and the cumulus scheme (Cumulus) were selected. They are labeled as the P1 scheme assemble using the YSU PBL scheme (Hong et al., 2006) along with the new SAS Cumulus scheme (Han and Pan, 155 2011), and the P2 scheme assemble using MYJ PBL scheme along with the BMJ Cumulus scheme. We choose these two sets of parameterization schemes mainly because according to our preliminary test (Figure S4) they had better performance than other scheme assembles in simulating the summertime precipitation in eastern China (which is an important indicator for model evaluation in this region). In addition, the comparison of these two assembles showed that the P1 scheme 160 generally produces a "warmer-and-wetter" climate conditions, while the P2 scheme generally leads to a "colder-and-dryer" climate (Figure S5, Figure S6). Therefore, the difference of these two simulation results can represent the impact of adjusting physical parameterization schemes to some extent. The spectral nudging was applied to the air temperature, wind components and relative humidity above the boundary layer in some experiments, with the coefficient of $3\times10^{-5}$ $s^{-1}$. The 165 nudging coefficient in our study was smaller than the default value in WRF ($3\times10^{-4}$ $s^{-1}$), in order to ensure the model variabilities (from physical schemes, external forcing, and other settings) can sufficiently developed and not be suppressed by large scale driving fields. Three type of external forcing configurations were applied to different experiments, including the time varying forcing, the climatological mean forcing, and no-varying forcing. By comparing these simulation results, we can 170 separate the influence of the forcing configuration, parameterization schemes and the spectral nudging on the long-term regional climate simulations in China.



## 3 Comparing the impacts of external forcing, spectral nudging, and physical parameterization schemes on long-term trend

The influence of model configurations on the long-term trends of annual mean temperature are shown in Figure 1. As shown by the observations, most regions in China had increasing temperature trend during 1961-2001, and the temperature increase were most significant in the northern part of China and the Tibet Plateau. The ERA40 reanalysis underestimated the magnitude of temperature trends in most regions. It's worth noting that the RCM could simulate the basic
warming trend even if the extern forcing is excluded in RCM because of the change of large-scale driving field (i.e. lateral boundary). When comparing the results of P1-GHG, P1-AERO, and P1-VOLC, we found the P1-GHG had most significant increasing temperature trend, followed by the P1-VOLC, and then the P1-AERO. It implies that using different kinds of time-varying forcing inside the RCM can strengthen or weakening the increasing temperature trend, perhaps mainly
depends on the sign of radiative forcing imposed by the forcing component. The P1-CTRL had stronger increasing trend in northern EC than the P2-CTRL, indicating that the physical parameterization schemes can also influence the long-term temperature trend in regional scales. Finally, the results of two simulations not using spectral nudging both had stronger trends at country scale, although they had some discrepancies in some areas. For instance, the temperature
trend of P1-GHG in northern EC was larger than that in P1-GHG-NNG, and P2-NNG had opposite trend in southern EC comparing with P2-CTRL. It suggests that the large-scale driving field can more significantly contribute to the increasing temperature trend in EC region, and using spectral nudging can improve the temperature trend simulation result in this region. However, better results were found in western China in the two no-nudging runs, implying the constrain of
large scale driving field had worsen the simulation results, possibly due to the sparse observations of western China in ERA40 data. The difference between P1-GHG-NNG and P2-NNG suggests that the influence of nudging on temperature trend may also depends on the parameterization schemes and external forcing inside the RCM.

        According to CN05.1 and CRU, for the eastern part of China, there was decreasing
precipitation trend in North China and increasing trend in south China, although the trends were not significant in most area (Figure 2). It has been known as the "North-dry-south-wet" pattern (NDSW). In northern China, opposite precipitation trend was found on the two side of the 105°E meridian line, with wetting trend to the west and drying trend to the east. This pattern is usually called the "West-wet-east-dry" pattern (WWED). The ERA40 generally reproduced the WWED
pattern but failed to reproduce the wetting trend in southern China. In addition, the reanalysis significantly overestimated the drying trend in southwestern China. The P1 simulations were similar with each other, showing a NDSW pattern in EC but wide range of drying trend to the west of 105 °E. These similar results implying the forcing configurations inside the RCM may had little effect on the precipitation trend. Among these simulations, the P1-GHG-NNG had largest area of
wetting trend in southern China. The P2-CTRL simulation had drying trend in most of China, suggesting the MYJ-BMJ scheme with spectral nudging cannot simulate the long-term precipitation trend in China correctly. However, when disabled the spectral nudging, the P2-NNG had completely opposite trend in southern China comparing to P2-CTRL. In fact, P2-NNG has the strongest wetting trend in southern China as well as the strongest drying trend in western China
among all the simulations. Comparing the simulations with or without nudging, we found that the spectral nudging had most significant impact on the precipitation trend, but just as the temperature



trend, the effect of nudging also depends on the selection of physical parameterization schemes.

We used the following method to separate the impact of changing external forcing, using spectral nudging, and changing parameterization schemes on the RCM simulated long-term trends in Eastern China (Table 2). Except for P1-GHG-NNG, the only difference among the six experiments using P1 physical schemes was the external forcing configuration inside the RCM. Therefore, the range of trend differences among these six P1 runs could be treated as the influence of changing external forcing on climatic trends. The impact of using spectral nudging on long-term trend was measured by the maximum trend difference among P1-GHG minus P1-GHG-NNG and P2-CTRL minus P2-NNG. Finally, the effect of physical schemes on climatic trend was represented by the absolute trend difference of P1-CTRL minus P2-CTRL.

As shown in Table 2, the annual temperature trend of CN05.1 during 1961-2001 in EC was 0.17 °C/10a, which manifested as a significant warming trend. The trend of CRU was slightly larger than CN05.1 (0.20 °C/10a), while the ERA40 reanalysis had smaller trend relative to the observations (0.13°C/10a). Most of the RCM runs reproduced the significant warming trend in EC, except for P1-AERO and P2-NNG. External forcing was also an important factor, which generated a fluctuation range of 0.08 °C/10a (0.12~0.20 °C/10a). The P1-CTRL and P2-CTRL simulated the same increasing temperature trend (0.16 °C/10a), implying the selection of physical parameterization schemes had little impact on the annual temperature trend in EC. The summer temperature showed a weak increasing trend in EC (0.02 °C/10a). Similar with the annual temperature trend, the trend variation range of three factors in a descending order was spectral nudging (0.24 °C/10a), external forcing (0.05 °C/10a), and physical parameterization schemes (0.03 °C/10a). With regard to winter temperature trend, however, the observations, reanalysis, as well as the RCM simulations all showed significant warming trend (more than twofold of the annual temperature trend). The three influencing factors had similar impact levels, with the contribution of trend variations were 0.1 °C/10a, 0.1 °C/10a, and 0.08 °C/10a for spectral nudging, external forcing, and physical parameterization schemes, respectively.

The annual precipitation during 1961-2001 for CN05.1 in EC region showed a weak increasing trend of 3.1 mm/10a. The annual precipitation trend of CRU was slight negative with the value of -1.1 mm/10a. The ERA40 reanalysis showed stronger negative trend (-20.1 mm/10a) relative to the observations, although still not significant. So, the large uncertainty exists in annual precipitation over EC region. The precipitation trend among RCM simulations showed large discrepancies, ranging from significant decreasing trend to significant increasing trend. Applying the spectral nudging to the P2 schemes caused the most significant variations of precipitation trend (63.4 mm/10a). Changing external forcing can lead to a trend variation of 19.9 mm/10a (-7.2~12.7 mm/10a), which was the second important factor. The selection of parameterization schemes caused a trend variation of 13.5 mm/10a. Note that although different adjustments in RCM produced different variation range of precipitation trend, all the variation range was larger than the real trend in EC (3.1 mm/10a). In other words, much cautions should be paid when dealing with the precipitation trend simulations because each single setting can reverse the sign and change the magnitude of precipitation trend, and the superposition effect may result in even larger discrepancies. The long-term trend of summer precipitation in EC was 9.5 mm/10a, slightly larger than the trend of annual precipitation but was not significant, either. All the RCM runs reproduced the positive trend except for the P2-CTRL. The dominant factor affecting the summer precipitation trend was the utilization of spectral nudging, which caused 38.9 mm/10a trend



variations. The difference of parameterization schemes can lead to a trend variation of 13 mm/10a, also regarded as an important impact factor. The influence of changing external forcing on the summer precipitation trend was relatively not obvious, since its variation range was 8.2 mm/10a, which was smaller than the real trend (9.5mm/10a). The winter precipitation trend was most

significantly affected by changing of external forcing, followed by physical parameterization scheme adjustments and enable spectral nudging, and their contribution of trend variation ranges were 1.8 mm/10a, 1.1 mm/10a, and 0.9 mm/10a, respectively.

## 4 Untangling the effect of single forcing

In this section, we further evaluated the influence of each single forcing components on regional climate simulation of China. We use the difference between dynamical single forcing runs (e.g., P1-GHG, P1-AERO, P1-VOLC) and climatological mean forcing run (P1-MFC) to separate the impact of individual forcing component.

### 4.1 GHGs and anthropogenic aerosols


We first evaluate the impact of forcing configurations on the simulation results of annual temperature trend. The dynamical all-forcing could lead to significant increasing trend in northern China and Tibet Plateau, and significant decreasing trend in central and southern China (Figure 3a). The varying GHGs forcing generated increasing temperature trend in most region, and

generally became stronger along with higher latitude, especially in northern and central China where the trends were significant (Figure 3c). On the contrary, the varying aerosols produced significant decreasing trend in most region of China, especially in southern China (Figure 3d). This uneven distribution of aerosol imposed temperature trend pattern was more similar with the trend pattern of water friendly aerosol other than ice friendly aerosol (Figure S2), indicating that

the anthropogenic aerosol was the most important components in explaining the decreasing temperature trend in China among all kinds of aerosols. Overall, the annual temperature increments in China (EC) during the simulation period caused by GHGs and aerosols were approximately 0.08 °C (0.07 °C) and -0.18 °C (-0.3 °C), respectively (Figure S7). In addition, the temperature trend pattern caused by dynamical all-forcing components was generally similar with

the linear superposition of GHGs and aerosols related trend patterns (Figure 3b), suggesting the most important forcing components for temperature trend in China were GHGs and anthropogenic aerosols. The difference of Figure (b) and (a) is likely contributed by the change of ozone and land use/cover.

The annual temperature trend pattern in China was generally in consistent with the decadal

change pattern of downward radiation flux (Figure 4). The GHGs caused decadal increasing of downward radiation flux in the whole region, mainly due to the increasing of longwave radiation flux (not shown). In contrast, aerosols caused the decreasing of shortwave radiation flux in the whole country, especially in central and southern EC. The decreasing shortwave radiation surpassed the increasing of longwave radiation in central and southern EC, while the opposite

situation was found in western part of China, thus resulting in the opposite temperature trend among these regions (Figure 3a). However, the temperature trend was not only affected by the radiative forcing, but also influenced by the dynamical responses. For instance, in northern part of EC and northeastern China, the temperature trend was positive, but the downward radiation flux in these regions showed decreasing trend. Further study revealed that there was an anti-cyclonic



circulation anomaly at lower levels along with positive geopotential height anomaly at middle level in northeastern China (not shown). Therefore, the increasing temperature in northern EC and northeastern China were mainly driven by adiabatic subsidence warming other than radiative forcing.

The forcing components could also influence the summer precipitation trend in China (annual
precipitation trend pattern was similar with summer precipitation trend pattern, not shown). The varying full set of forcing caused decreasing precipitation trend in EC, northeastern China, and southwestern China, and the significant decreasing trend were mainly distributed in North China and southwestern China with their local impact larger than 20 mm/10a (Figure 5a). However, the precipitation trend imposed by varying GHGs had a completely different pattern, showing a
generally increasing trend at country scale and a highly discrete distribution pattern at local scale (Figure 5c). The varying aerosols had resulted in decreasing precipitation trend in most regions, which was similar with the all-forcing pattern (Figure 5d). The annual precipitation changes in China (EC) during the simulation period caused by GHGs and aerosols were approximately 16.4 mm (13.3 mm) and -44.3 mm (-69.3 mm), respectively (Figure S7). However, unlike the
temperature trend, the linear superposition precipitation trend pattern of GHGs and aerosols showed lower similarity to the all-forcing pattern in comparison with the aerosol-only pattern (Figure 5b). It implies the aerosols, especially the anthropogenic aerosols, were the dominant forcing components affecting the summer precipitation trend in China.

According to previous studies, GHGs can modulate the summer precipitation in China
mainly through two mechanisms. Firstly, due to the different thermal capacity between land and sea surface, the global warming will increase the land-sea thermal contrast and thus lead to stronger monsoon circulations. Stronger monsoon circulation (mainly the strengthen of southerlies) will shift the summer rain belt more northward, and favors the north-wet-south-dry precipitation anomaly pattern in Eastern China (e.g., He and Zhou, 2020; Seth et al., 2019). Secondly,
according to the law of Clausius-Clabeyron, more GHGs lead to higher mean temperature and thus result in more water vapor content in the atmosphere, which favors the formation of precipitation (Chen et al., 2020b; Zhou et al., 2009). Anthropogenic aerosols mainly affect precipitation in monsoon regions through its negative radiative forcing effect. For instance, the aerosol loading is generally much larger above land areas than above the sea, which reduces the
land-sea thermal contrast and weakens the monsoon circulation, and thus lead to less monsoon precipitation (Bollasina et al., 2011; Wilcox et al., 2013).

To further explain the precipitation responses to external forcing, we analyzed the low-level atmospheric circulation and specific humidity change between 1981~2000 and 1961~1980 (Figure 6). For the responses to GHGs (Figure 6c), most area of EC was dominated by a southwesterly
wind anomaly, which indicates the strengthening of the East Asian Summer Monsoon (EASM). Stronger southerlies brought more water vapor to the northern EC and reduced the water vapor in southeastern EC, which were generally consistent with the precipitation trends among these regions (Figure 5c). However, the more "noisy" distributed precipitation trend pattern than the circulation and water vapor anomaly pattern suggested other random processes might also
contributed to the precipitation responses, such as the changes of convective precipitation. In general, the WRF model's summer precipitation changes responses to GHGs seems mainly determined by dynamical processes. On the other hand, the aerosol-related pattern showed an anti-cyclonic anomaly centered in the coastal region of northern EC, accompanied by the



decreasing of low-level water vapor content (Figure 6d). In southern part of EC, the low-level was mainly dominated by the northerly wind anomaly, suggesting the weakening of the EASM. The anti-cyclone and weakening EASM reduced the low-level water vapor transport north to the Yangtze River, resulting in the vast decreasing trend among the EC region (Figure 5d). The circulation anomaly pattern of all-forcing components (Figure 6a) was generally more like the aerosol's pattern other than the linear superposition pattern of GHGs and aerosols (Figure 6b), which confirmed the dominant role of aerosols in explaining the circulation responses to external forcing. Overall, considering the patterns of temperature, precipitation, and other physical variables, the WRF model's responses to GHGs and anthropogenic aerosols seems reasonable.

### 4.2 Volcanic aerosols

We used the temperature difference between P1-VOLC and P1-MFC in summer to represent the impact of dynamical volcanic aerosols, because the most significant responses to volcanic aerosols was usually the decreasing of summer temperature. During our study period, there were five volcanic eruptions could be recognized in total, and the eruptions in 1963, 1982, and 1991 were obviously stronger than the climatological mean level (Figure 7a). The summer temperature of P1-VOLC in three large eruption years were generally lower than the result of P1-MFC, implying the influence of dynamical volcanic aerosols could be identified (Figure 7b). The temperature declines after these three large eruptions were more obvious in EC (up to -0.1 ℃) than in the whole china (up to -0.05 ℃). However, there were also obvious temperature differences in no volcanic eruption years, with maximum temperature differences up to 0.15 ℃ for EC and 0.1 ℃ for China, respectively. These differences might come from the random variability of WRF model. It means the impact of using dynamical volcanic aerosol forcing was weaker than the model random variability. Therefore, it seems that there were no significant differences between using dynamical volcanic forcing and using climatological mean volcanic forcing.

### 5 Discussions

Jerez et al. (2018) proposed that using dynamical GHGs in RCMs can resulted in 1~2K overestimate of warming signal at century scale in the whole European region, and in some local areas the warming signal may even double. They considered only GHGs forcing, and disabled the nudging option while simulating. In our case, the annul temperature trend of GHGs forcing run (P1-GHG) was 0.04 ℃/10a larger than the all-forcing run (P1-CTRL) in EC region (see Table 2), and the latter was closer to the observation. It suggests that using all set of forcing can improve the temperature trend simulation in EC, probably mainly because the anthropogenic aerosols' cooling effect can offset the overestimation of warming signal when using GHGs only. We also found that the utilization of spectral nudging had more significant impact both on temperature trend and precipitation trend than the configuration of external forcing. Furthermore, the influence of nudging seems also depends on the selection of physical parameterization schemes. Therefore, we think the following suggestions can help improving the simulation results and reducing the uncertainties when conducting long-term dynamical downscaling simulations. First, the optimal parameterization schemes can be obtained by conducting a group of sensitivity test. This sensitivity test can be performed on a short-term simulation period for saving computation resources, such as one year, which mainly focus on the model's ability for spatial patterns (such as the temperature distribution, the position of main rain belt). Second, we strongly recommend that



at least two long-term tests should be performed before the formal simulation to determine whether to use nudging or not. Third, use all set of dynamical forcing components rather than use GHGs forcing only, especially in the strong anthropogenic aerosol source regions. Finally, we agreed with Jerez et al. (2018) that one should clearly records the RCMs configurations for comparisons and repetitions.

Our work has supplemented the forcing components in the WRF model, and the relevant analysis has shown that the WRF model's responses to dynamical forcing are reasonable. However, there are still some uncertainties in our work. The dynamical aerosols in our work did not consider the black carbon, which is recognized as an important forcing component with positive radiative forcing effect at surface. Therefore, the warming signal in our work might be underestimated to some extent. We used same nudging coefficients in all the nudging simulations, and the impact of nudging coefficients need further studies. Other factors we did not consider may also influence the model' responses to forcing, such as domain, resolution, and time period.

## 6 Conclusions

We modified the WRF model to include the spatial-temporal varying external forcing components and compared the influence of using varying forcing on the long-term trend simulations to other model configurations. The impacts of single forcing components are also separated and discussed. The main results are listed below:

1) Using different forcing configurations inside the RCMs could resulted in approximately 0.08 ℃/10a variations for annual temperature trend and 19.9 mm/10a variations for annual precipitation trend in EC, which was stronger than the influence of parameterization schemes but was weaker than the impact of spectral nudging. When dealing with summer precipitation trend or winter temperature trend simulations, much attention should be paid because all the three factors discussed in this paper would have considerable impact on the trend magnitude or even sign. The influence of spectral nudging on long-term trends also depended on the selection of parameterization schemes and forcing components. We suggest that when using RCMs for long-term simulations, a preliminary test is needed to decide whether or not to use nudging. Generally it is really relevant with the large-scale driving field and the research aeras. One should also use all set of varying forcing components instead of using varying GHGs only.

2) Among all the forcing components, GHGs and anthropogenic aerosols had the largest impact on annual temperature trend in China, and the forcing response pattern could be roughly explained by the linear superposition of these two forcing components. Anthropogenic aerosols had most significant impact on the trend pattern of summer precipitation. During the simulation period, the annual temperature increments in China (EC) caused by GHGs and anthropogenic aerosols were approximately 0.08 ℃ (0.07 ℃) and -0.18 ℃ (-0.3 ℃), respectively, while the annual precipitation increments were approximately 16.4 mm (13.3 mm) and -44.3 mm (-69.3 mm), respectively. The impact of volcanic aerosols on summer temperature change was up to 0.05 ℃ (0.1 ℃) in China (EC), but seemed not strong enough to be separated from the random variabilities of WRF model. The spatial pattern responses to dynamical external forcing of WRF model were generally reasonable.



### Code and Data availability

The exact version of the model used to produce the results used in this paper is archived on Zenodo (Feng Jinming. (2023). WRF v3.8.1 with external forcing components (v3.8.1). Zenodo. https://doi.org/10.5281/zenodo.8111877, last access: 4 July 2023). The CRU TS4.01 dataset is available at https://crudata.uea.ac.uk/cru/data/hrg/cru_ts_4.01/cruts.1709081022.v4.01/ (last access: 4 July 2023). The ERA40 reanalysis data is available at https://apps.ecmwf.int/datasets/data/era40-daily/levtype=sfc/ (last access: 4 July 2023). The external forcing data is available at https://pcmdi.llnl.gov/mips/cmip5/forcing.html (last access: 4 July 2023).

### Competing interests

The authors declare no conflicts of interests exits in this paper.

### Author contribution

J.M. Feng modified the WRF model and designed the experiments. M. Luo conducted the numerical simulations and wrote the manuscript. J. Wang and Y. Qiu contributed to the figures and analysis. Q.Z. Wu and K. Wang provided advices for the analysis and manuscript.

### Acknowledgement

This work was supported by the General Project of the National Natural Science Foundation of China (42275186), the Basic Research Project of the Yunnan Province Science and Technology Department (202201AU070196), the National Key R&D Program of China (2016YFA0600403), the Key R&D Plan of Yunnan Province Science and Technology Department-Social Development Special Project (202203AC100005), the innovation team "Greenhouse gas and carbon neutral monitoring and evaluations" of Yunnan Meteorological Service (2022CX05), and the Natural Science Foundation of Yunnan Province (202302AN360006). The work was carried out at the National Supercomputer Center in Tianjin, and the calculations were performed on TianHe-1 (A).

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









**Figures and Tables**

**Table 1** Sensitivity tests of various factors which affects the RCM climate simulations

| Experiment name | PBL and Cumulus scheme | Use spectral nudging or not | External forcing configuration |
|---|---|---|---|
| P1-CTRL | YSU-newSAS | Yes | Varying all-forcing |
| P1-MFC | YSU-newSAS | Yes | Climatological mean all-forcing |
| P1-NFC | YSU-newSAS | Yes | No-varying forcing (GHGs, solar constant, land use fixed at 1958, no varying aerosols and ozone input) |
| P1-GHG | YSU-newSAS | Yes | Varying GHGs, climatological mean other forcing |
| P1-GHG-NNG | YSU-newSAS | No | Varying GHGs, climatological mean other forcing |
| P1-AERO | YSU-newSAS | Yes | Varying aerosols, climatological mean other forcing |
| P1-VOLC | YSU-newSAS | Yes | Varying volcanic aerosols, climatological mean other forcing |



| P2-CTRL | MYJ-BMJ | Yes | Varying all-forcing |
| P2-NNG | MYJ-BMJ | No | Varying all-forcing |

**Table 2** Long-term trend (1961–2001) of area weighted mean temperature and precipitation of EC region (105°–123°E, 20–45°N)

| Data source | Annual temperature (℃/10a) | JJA temperature (℃/10a) | DJF temperature (℃/10a) | Annual precipitation (mm/10a) | JJA precipitation (mm/10a) | DJF precipitation (mm/10a) |
| --- | --- | --- | --- | --- | --- | --- |
| CN05.1 | 0.17* | 0.02 | 0.38* | 3.1 | 9.3 | 5.2 |
| CRU | 0.2* | 0.05 | 0.41* | -1.1 | 8.2 | 4.2 |
| ERA40 | 0.13* | -0.01 | 0.35* | -20.1 | -0.4 | 3.5 |
| P1-CTRL | 0.16* | 0.07 | 0.38* | -7.2 | 6 | 1.8 |
| P1-MFC | 0.18* | 0.09 | 0.4* | 10.4 | 13.3 | 3.2 |
| P1-NFC | 0.19* | 0.08 | 0.41* | 6.6 | 11.3 | 2.9 |
| P1-GHG | 0.2* | 0.11* | 0.43* | 12.7 | 14.2 | 3.3 |
| P1-GHG-NNG | 0.19* | 0.03 | 0.53* | 31.6* | 27.8* | 4.2 |
| P1-AERO | 0.12 | 0.06 | 0.33* | -6.1 | 7.4 | 1.5 |
| P1-VOLC | 0.18* | 0.11* | 0.39* | 9.4 | 11.6 | 3 |
| P2-CTRL | 0.16* | 0.04 | 0.46* | -20.7* | -7 | 2.9 |
| P2-NNG | 0.05 | -0.2* | 0.52* | 42.7* | 31.9* | 3.5 |

*Note:* The values with * represent the trend was significant for the Mann-Kendall test under 0.05 significance level.


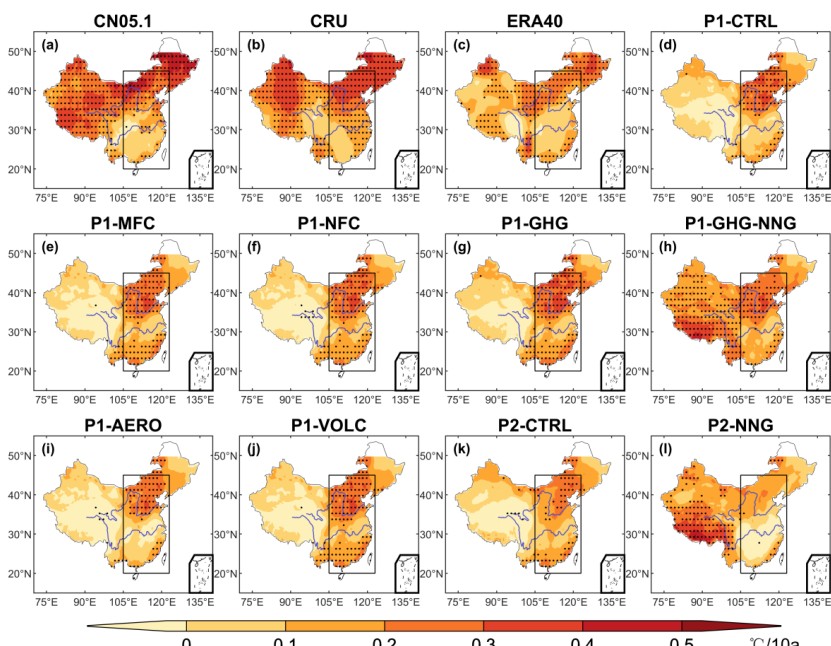

**Fig. 1** Long-term trend of annual mean temperature in China during 1961–2001. The trends in stippled regions are significant under 0.05 level by the Mann-Kendall test. Black rectangle is the EC region (105°–123°E, 20–45°N).


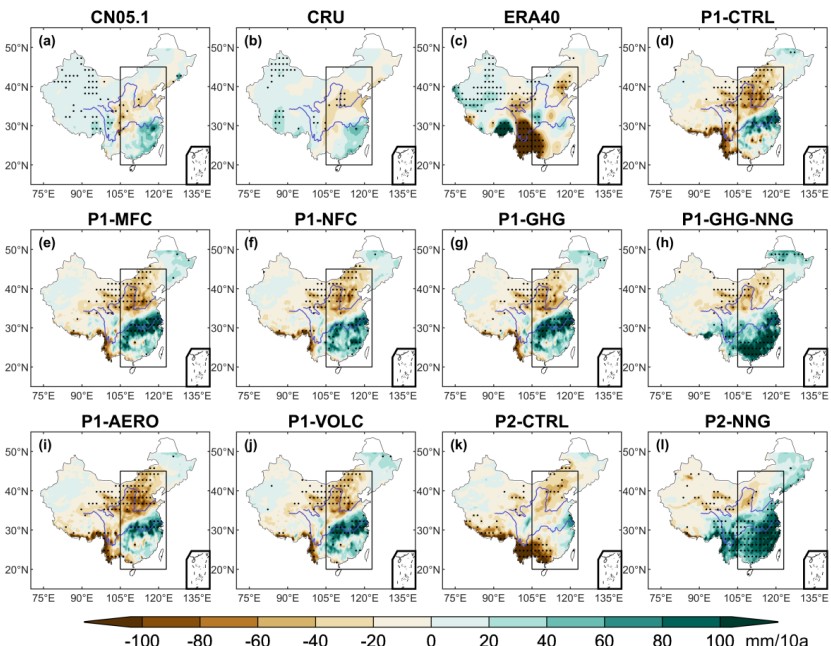

**Fig. 2** Long-term trend of annual accumulated precipitation in China during 1961–2001. The trends in stippled regions are significant under 0.05 level by the Mann-Kendall test. Black rectangle is the EC region (105°–123°E, 20–45°N).

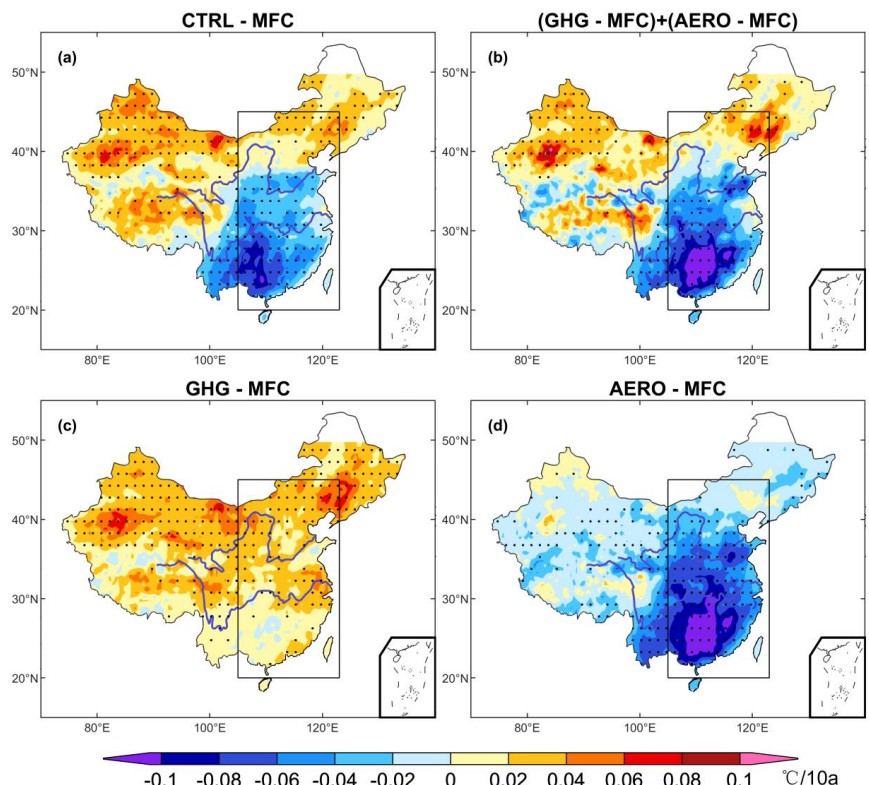

**Fig. 3** The annual temperature trend from 1961 to 2001 for P1-CTRL minus P1-MFC (a),
P1-GHG minus P1-MFC plus P1-AERO minus P1-MFC (b), P1-GHG minus P1-MFC (c), and
P1-AERO minus P1-MFC (d), respectively. Black rectangle is the EC region (105°–123°E, 20–
45°N).



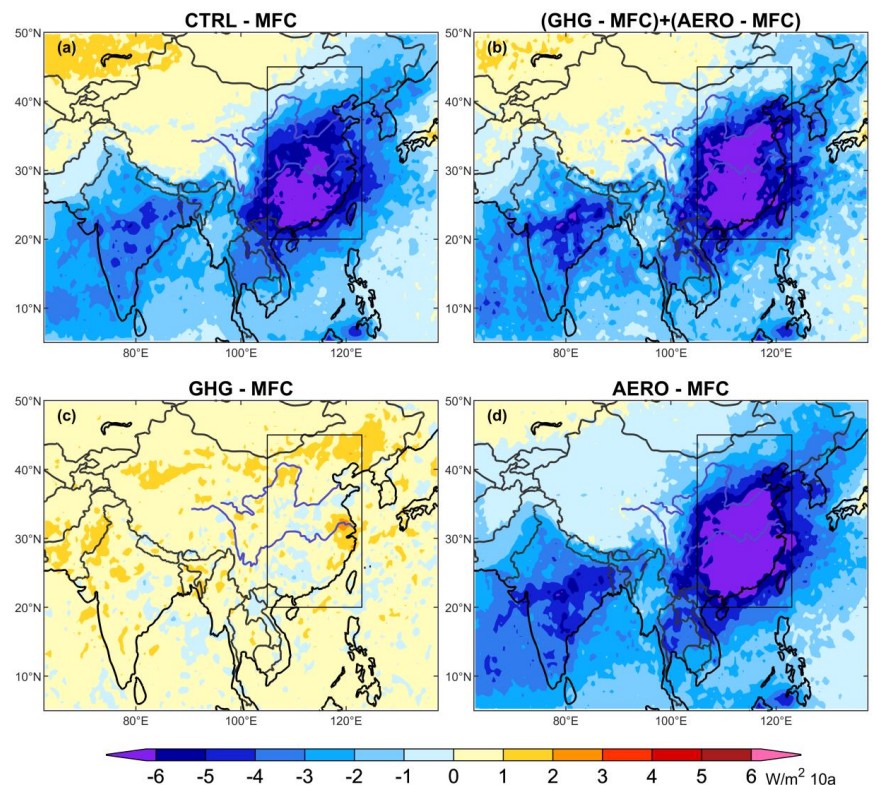

**Fig. 4** The difference of two climatological mean (1981~2000 minus 1961~1980) downward shortwave plus longwave radiation flux for P1-CTRL minus P1-MFC (a), P1-GHG minus P1-MFC plus P1-AERO minus P1-MFC (b), P1-GHG minus P1-MFC (c), and P1-AERO minus P1-MFC (d), respectively. Black rectangle is the EC region (105°–123°E, 20–45°N).

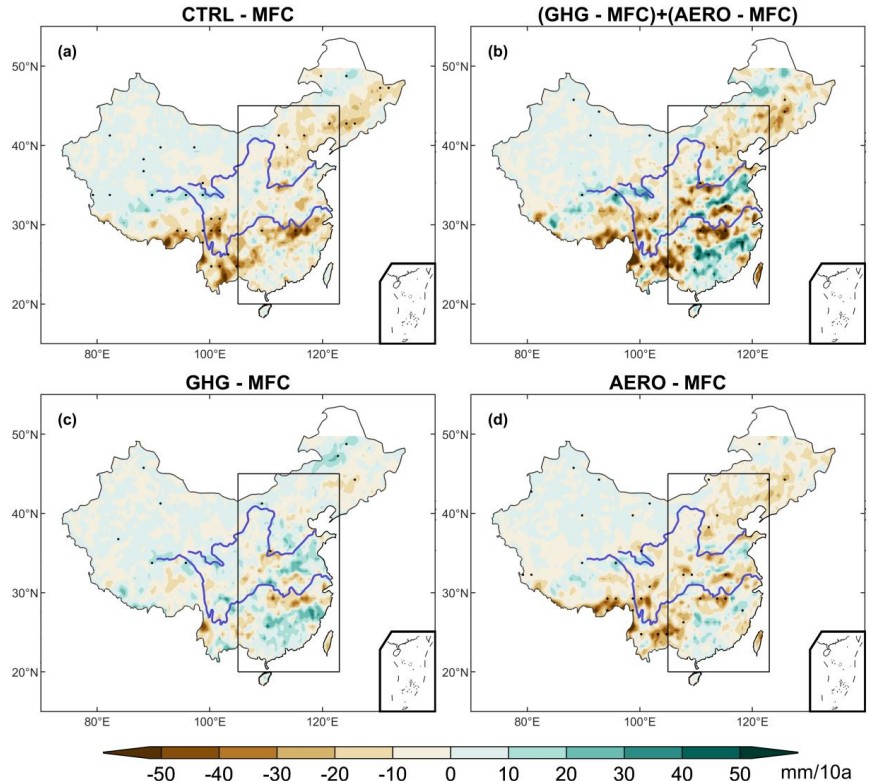

**Fig. 5** The summer precipitation trend from 1961 to 2001 for P1-CTRL minus P1-MFC (a), P1-GHG minus P1-MFC plus P1-AERO minus P1-MFC (b), P1-GHG minus P1-MFC (c), and P1-AERO minus P1-MFC (d), respectively. Black rectangle is the EC region (105°–123°E, 20–45°N).

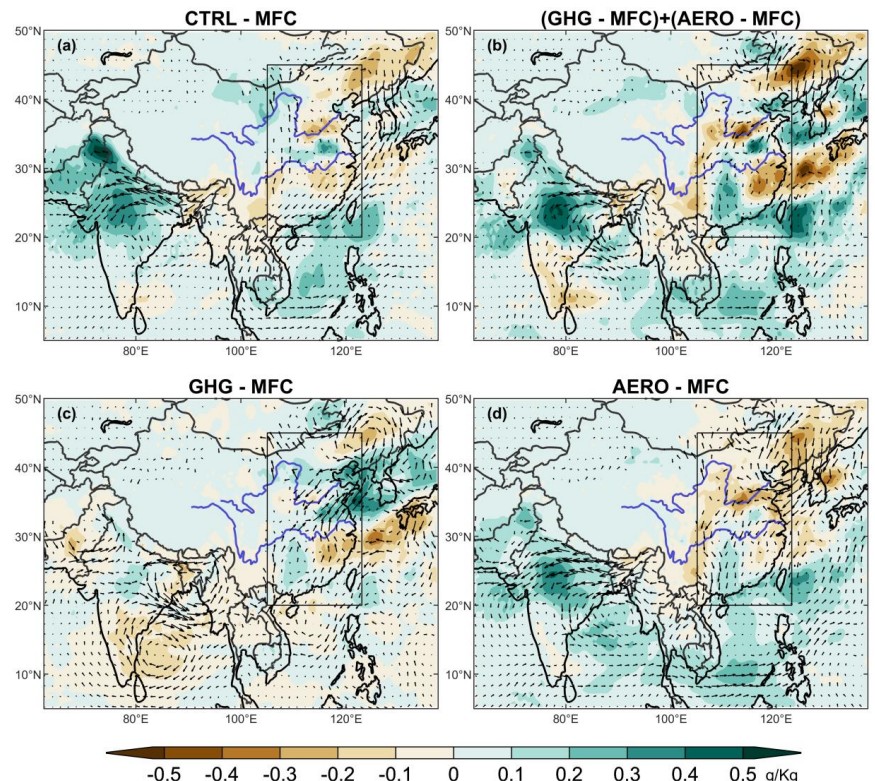

**Fig. 6** The difference of two climatological mean (1981~2000 minus 1961~1980) of summertime wind vector at 850 hPa (arrow) and vertical integrated specific humidity in lower levels (1000~600 hPa, filled color) for P1-CTRL minus P1-MFC (a), P1-GHG minus P1-MFC plus P1-AERO minus P1-MFC (b), P1-GHG minus P1-MFC (c), and P1-AERO minus P1-MFC (d), respectively. Black rectangle is the EC region (105°–123°E, 20–45°N).

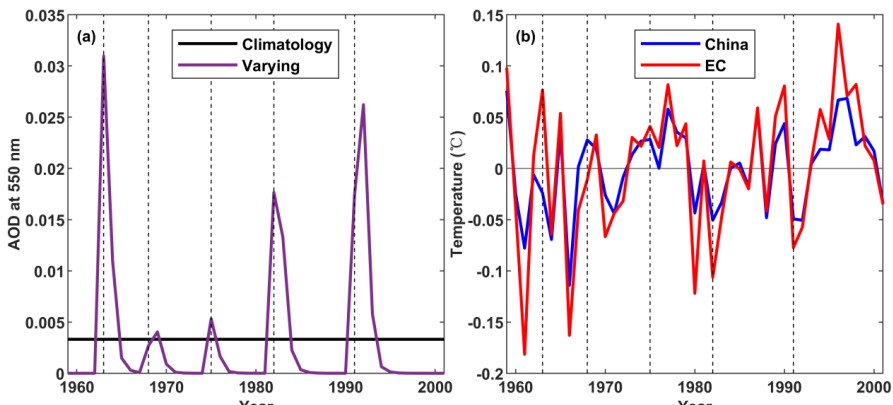

**Fig. 7** The time series of regional mean summertime vertical integrated volcanic aerosols (a) and temperature difference between P1-VOLC and P1-MFC experiments (b). Purple line and black line in (a) represent the varying volcanic aerosols and climatological mean volcanic aerosols, respectively (for the whole domain). Blue line and red line in (b) represent the temperature difference in China and in EC region, respectively. Black dash lines represent the volcanic eruption years.