# Peer review of "The influences of incorporating dynamical external forcing in WRF v3.8.1 on regional climate simulation in"

_EGUsphere, 2023_

## Referee Comment (RC1)

Review of "*The influences of incorporating dynamical external forcing in WRF v3.8.1 on regional climate simulation in China*"

By: Jinming Feng, Meng Luo, Jun Wang, Yuan Qiu, Qizhong Wu, and Ke Wang

This study investigates the influence of varying external climate forcings on regional climate simulations over China. The authors modified parts of the WRF v3.8.1 code to include varying external forcing for greenhouse gases, anthropogenic aerosols, and volcanic aerosols, instead of using the climatological means that are usually prescribed. The authors then compared simulations with climatological means for all forcing, varying all external forcing or varying one forcing component and keeping climatological means for the others. They also investigated the influence of nudging and the choice of parameterization schemes on these simulations. The authors recommended varying all external forcing components for regional climate model simulations as well as preforming long-term preliminary tests to determine whether to consider nudging or not.

In my opinion, the presented results lack robustness, owing to the limited number of simulations performed. Also, for a Development and Technical paper, it lacks enough technical details, especially on the specific changes made to the WRF code, the specific parameters/variables modified in the code, details about the nudging, etc. These needs to be addressed thoroughly before the paper is further considered for GMD.

**Main comments:**

My main concern, which is a fairly serious one, is the lack of robustness in the results presented. This stems from inconsistency in the experimental designs. There are 7 P1 simulations (the control and 6 experiments). But there are just 2 for P2 simulations. Even when we look at the P1 simulations only, there are many inconsistencies in the experimental designs. For example, there is a P1-GHG (with nudging) and a P1-GHG-NNG (without nudging). Why did the authors not use the same convention to design the other experiments? Consequently, the impact of nudging (for the P1 simulations) is effectively investigated only for P1-GHG but not the others. However, the influence of nudging could be different for different model configurations and choice of parameterization schemes (e.g., Song et al. (2011), Wootten et al. (2016)). Therefore, the results are thus not robust enough to be generalized the way they are presented in the paper.

In my opinion, further simulations are needed to investigate the robustness of the results presented. Otherwise, there should enough compelling reasons in the text to address the inconsistency in the experimental design, although this may not be enough to address the lack of robustness of the results.

Additionally, the lack of ensemble simulations for each configuration makes it impossible to investigate any influence of the model's internal climate variability. Separating the internal variability from the forced external signals could be essential to could also be key to understand the role of anthropogenic climate change in the reported trends of temperature and precipitation (Frankcombe et al. 2015).

**Specific comments:**

- The authors should mention the values used for the different nudging parameters.

- L69: change resulted to result.

- L70: remove quiet.

- L81: change resulted to result.

- L82: change region to regions.

- L84: "pay cautious" to "be cautious".

- L87: remove were

- L91: change are to is

- L91-93: Following from my main comment above: The authors said that "China is one of the most densely populated areas in the world, and its monsoon climate are strongly modulated by internal climate variabilities ….". If this is the case, why was there not any investigation with different ensemble members of the same configuration to investigate the impact of the internal variation on the results?

- L103: change focus to focusing

- L113-117: Here the authors start by saying "The present version of WRF has only considered ...." and the end by saying "...physics schemes in WRF v3.8.1". This makes one think WRF v3.8.1 is the current version of WRF, which is obviously not the case (I believe there is a WRF v4.5 currently). I think this needs to be clarified. It may also be necessary to mention why WRF v.3.8.1 is used rather than the current version. And if there are any differences, the authors should briefly describe those as well.

- L122-124: What is the criteria for considering water-friendly aerosols and ice-friendly aerosols as anthropogenic and natural aerosols respectively? If there any studies that has shown this, please cite them.

- L133: Please give examples of studies that have used those two datasets for research and model evaluation.

- L159-161: This is only true for the southeastern part of the domain. I think it should be mentioned.

- L180: external

- L184: weaken

- L185-187: This is very well known already. Different parameterization schemes can considerably alter simulation results.

- L189-193: Here the authors suggest that "spectral nudging can improve temperature trend simulation…". In my opinion, this is not robust since the authors did not investigate it for the other configurations.

- L196-198: This will be clearer/robust if there is a P2-GHG and a P2-GHG-NNG as well.

- L208-209: I am not sure if this is entirely true. In my opinion, this is true when considering only the sign of the trend but not the magnitude. I think the forcing configurations can have a considerable localized influence on the magnitude of the trend. If you zoom into the southern areas of the box shown on the P1 figures, you can notice some changes in the magnitudes of the trend. This should be clarified.

- L232-234: A robust conclusion cannot be made based on just the 2 choices used here. I suggest the authors revise this so that it does not seem like this result is generalizable.

- L246: change "So, the large…" to "So, a large…"

- L254: cautions to caution

- L250: (63.4 mm/10a) Is this value reported in Table 2? Because I cannot find it. I see -20.7 for P2-CTRL. Please check and clarify.

- L256-257: The authors should make it clear which data source is being spoken about here. Is it the observations? If yes, which one? They should also ensure that the values reported in the text correspond with those in the Table.

- L279: region to regions

- L294: remove in

- L330: Did the authors mean Clausius-Clapeyron?

- L339: area to areas

- L345: contributed to contribute

- L362-364: I think this sentence should be revised, and possibly broken down into two, as it is a bit difficult to understand.

- L364: remove obviously

- L370-371: The reason why there needs to be more ensemble members to evaluate internal variability. This should be addressed somehow in the text at least.

- L377: change overestimate to overestimation

- L386: remove seems

- L396: change records to record

- L412: change resulted to result

**References**

Frankcombe, L. M., England, M. H., Mann, M. E., & Steinman, B. A. (2015). Separating internal variability from the externally forced climate response. *Journal of Climate*, *28*(20), 8184-8202. https://doi.org/10.1175/JCLI-D-15-0069.1

Song, S., Tang, J. & Chen, X. Impacts of spectral nudging on the sensitivity of a regional climate model to convective parameterizations in East Asia. *Acta Meteorol Sin* **25**, 63–77 (2011). https://doi.org/10.1007/s13351-011-0005-z

Wootten, A., Bowden, J. H., Boyles, R., & Terando, A. (2016). The sensitivity of WRF downscaled precipitation in Puerto Rico to cumulus parameterization and interior grid nudging. *Journal of Applied Meteorology and Climatology*, *55*(10), 2263-2281. https://doi.org/10.1175/JAMC-D-16-0121.1

---

## Referee Comment (RC2)

**"The influences of incorporating dynamical external forcing in WRF v3.8.1 on regional climate simulations in China" by Feng et al.,**

The authors modified the WRF v3.8.1 to include the spatial-temporal varying external forcing components to study the impact of dynamical forcing on the long-term simulation in China. Though the objective of the work is fascinating, I have some serious concerns about the WRF modifications and the experimental design.

1. One of the major points of this work is the development of WRF to include the external forcing components in WRF. But authors spent just one paragraph (Model improvement) under the 'Model and Data' section, which is also very general. They did not spend a single sentence about their improvements compared to the current version of the WRF (WRF v4.5.1). Why did the authors modify the WRF v3.8.1 is also unclear and explained in the manuscript? WRF v3.8.1 was released in Aug 2016, so why we need to improve that 7-year-old version compared to the current version is also not explained.

2. My second primary concern of this manuscript is its experimental design. The authors recommended judging the need for the nudging technique before the formal simulation, but their experimental design for nudging is surprising to me. Under the P1 subset, authors performed a control simulation, followed by all forcings and individual forcings of GHGs, Aerosol, and Volcano. All the experiments mentioned above were performed using Nudging techniques, but suddenly then, they performed only a single simulation without Nudging experiments (for GHGs). What is the scientific basis for choosing GHGs forcing without Nudging experiments? Why not choose all forcing experiments for Nudging justification? I recommend that authors perform without Nudging experiments, at least for all forcings and Aerosol forcings on top of GHG forcings. Also, for the P2 experiment, they only performed a control simulation and nudging experiment, but their major objective was to see the impact of external forcings.

3. Panel 'b' in Figures 3 – 6 shows the combined effects of GHG and Aerosol on the temperature, precipitation, and circulation trend. I want another experiment by adding GHG and Aerosol as external forcings. I believe that impact might have differed from panel 'b' in Figures 3 – 6 since they are not in a linear relationship in the actual

atmosphere. We might not speculate some conclusions by linearly adding to independent single forcing simulations specifically for precipitation.

My other points are also listed below.

4. This point is related to my first major concern. How is your modification different from the most recent WRF version (WRF 4.5.1) regarding aerosol contribution through Thompson Microphysics and RRTMG radiation scheme?

5. Authors must provide more detail about their modification and sensitivity with and without modification experiments, at least for a single sensitivity.

6. Why are authors using the ERA40 reanalysis, not the most recent ERA5, to force the WRF model? ERA5 provides much better spatial and temporal resolution compared to ERA40.

7. Similar to the above point, why authors are using CMIP5 forcings instead of CMIP6 data? These can make a significant difference in their simulations.

8. Figure S4 is very much ambiguous, and it's difficult to quantify which scheme performs better other than by eye estimation. Authors should do a more quantitative way to choose the best scheme, and authors can try the Taylor Diagram for U, Geopotential, Temperature, and Precipitation.

9. To quantify the Nudging effect robustly, authors must perform without nudging experiments for all forcings (P-MFC-NNG) and aerosol forcing (P1-AERO-NNG). This will help them to quantify the nudging impact.

10. For precipitation trend analysis, how reliable are CRU and ERA40 precipitation data?

---

## Author Comment (AC1)

**REPLIES TO EDITORS' AND REFEREES' COMMENTS**

**We sincerely appreciate editor's and referees' highly constructive suggestions and comments, upon which we have revised our manuscript. Point-by-point replies are as follows with referees' comments in black and our replies in blue. In the revised manuscript, all the revisions are highlighted in blue as well.**

**Referee 1:**

Comments to the Author

This study investigates the influence of varying external climate forcings on regional climate simulations over China. The authors modified parts of the WRF v3.8.1 code to include varying external forcing for greenhouse gases, anthropogenic aerosols, and volcanic aerosols, instead of using the climatological means that are usually prescribed. The authors then compared simulations with climatological means for all forcing, varying all external forcing or varying one forcing component and keeping climatological means for the others. They also investigated the influence of nudging and the choice of parameterization schemes on these simulations. The authors recommended varying all external forcing components for regional climate model simulations as well as preforming long-term preliminary tests to determine whether to consider nudging or not. In my opinion, the presented results lack robustness, owing to the limited number of simulations performed. Also, for a Development and Technical paper, it lacks enough technical details, especially on the specific changes made to the WRF code, the specific parameters/variables modified in the code, details about the nudging, etc. These needs to be addressed thoroughly before the paper is further considered for GMD.

**Main comments:**

1)My main concern, which is a fairly serious one, is the lack of robustness in the results presented. This stems from inconsistency in the experimental designs. There are 7 P1 simulations (the control and 6 experiments). But there are just 2 for P2 simulations. Even when we look at the P1 simulations only, there are many inconsistencies in the experimental designs. For example, there is a P1-GHG (with nudging) and a P1-GHG-NNG (without nudging). Why did the authors not use the same convention to design the other experiments? Consequently, the impact of nudging (for the P1 simulations) is effectively investigated only for P1-GHG but not the others. However, the influence of nudging could be different for different model configurations and choice of parameterization schemes (e.g., Song et al. (2011), Wootten et al. (2016)). Therefore, the results are thus not robust enough to be generalized the way they are presented in the paper.

In my opinion, further simulations are needed to investigate the robustness of the results presented. Otherwise, there should enough compelling reasons in the text to address the inconsistency in the experimental design, although this may not be enough to address the lack of robustness of the results.

Reply: Thank you for the constructive comments. Indeed, the experiments are not strictly consistent for different configurations. The main reasons for the inconsistency are as follows:

1) The fully consistent experiment is too expensive. Currently we have conducted 9 climate simulations. But if we try to strictly fulfill the consistency in experiment design, we must pay several fold of computing resources for this work, and this may be unaffordable.

2) The current experiment design is not random or unfounded. First, we did not set a simulation of P1-NFC-NNG (using P1 physical schemes, no forcing in the RCM, no spectral nudging), because this is a common choice for most RCMs users (it means most researchers do not use dynamical external forcing, maybe some of them use spectral nudging, and their choices for physical schemes are not limited for the choice of this paper). Second, we compared the effect of spectral nudging for P1-GHG, because GHGs forcing is added to WRF model since version 3.5, the WRF users can turn on this option without the modification of WRF model. Therefore, the combination of spectral nudging and GHGs forcing is a potential choice for all the WRF users, and currently such evaluations is insufficient, so we did this thing. Another reason is GHGs are the most importance external forcing, and other forcing may have contradict effect to GHGs such as aerosols. What we want to know is the potential of the maximum difference between the simulations of using nudging and not using nudging, and focusing on the most important forcing is our best choice. Third, we also compared the influence of spectral nudging for P2-CTRL, because this is the situation of considering all the forcing components. In theory, using all kinds of forcing is better than using a single forcing. Currently there are rarely works have discussed the usage of nudging will change the simulation result to what extend when using all kinds of forcing in RCM. Although the simulations here are far from sufficient enough, we think the comparison between P2-CTRL and P2-NNG is also beneficial to give a preliminary answer to this question. We did not expect to test the influence of nudging on all the situations (and this is also unrealistic), as the main purpose of this paper is to evaluate the impact of adding dynamical external forcing in the RCM.

We also reorganized the 2.3 experiment design section, and explained the reason for the current experiment design. Please see the updated manuscript.

2) Additionally, the lack of ensemble simulations for each configuration makes it impossible to investigate any influence of the model's internal climate variability. Separating the internal variability from the forced external signals could be essential to could also be key to understand the role of anthropogenic climate change in the reported trends of temperature and precipitation (Frankcombe et al. 2015).

Reply: Thank you for the comments. The influence of internal climate variability should be treated cautiously when using Global coupled climate models (GCMs).

When running the GCMs, only initial conditions are provide to the model, so any little differences in the initial fields can resulted in significant differences as the integration continues. Therefore, conducting ensemble simulations is a conventional way for GCMs to evaluate the uncertainty of model internal variabilities. However, for the RCMs, this is not so much important, because the RCMs are further restrained by the lateral boundary conditions. Generally, the influence of initial condition can maintain for several years at most, and as the integration time increased the lateral boundary conditions are far more important than the initial conditions. So the differences between ensemble members of RCMs are much smaller than of the GCMs. In order to save computing resources, we did not performed the ensemble simulations.

**Specific comments:**

1) - The authors should mention the values used for the different nudging parameters.
Reply: We applied the same nudging coefficients $3\times10^{-5}$ s$^{-1}$ to the different nudging parameters, please see L163-L167 in the manuscript.

2) - L69: change resulted to result.
Reply: The word "resulted" has been changed to "result".

3) - L70: remove quiet.
Reply: The word "quite" has been removed.

4) - L81: change resulted to result.
Reply: The word "resulted" has been changed to "result".

5) - L82: change region to regions.
Reply: The word "region" has been changed to "regions".

6) - L84: "pay cautious" to "be cautious".
Reply: The word "pay" have been changed to "be".

7) - L87: remove were
Reply: The word "were" has been removed.

8) - L91: change are to is
Reply: The word "are" has been changed to "is".

9) - L91-93: Following from my main comment above: The authors said that "China is one of the most densely populated areas in the world, and its monsoon climate are strongly modulated by internal climate variabilities ….". If this is the case, why was there not any investigation with different ensemble members of the same configuration to investigate the impact of the internal variation on the results?

Reply: Thank you for the comments. Same as the replies for main comments. The lateral boundary conditions will significantly suppress the internal variability in the RCMs. Therefore, we did not perform the ensemble simulations to save computing resources.

10) - L103: change focus to focusing
Reply: Thank you for reminding. We revised this word.

11) - L113-117: Here the authors start by saying "The present version of WRF has only considered ...." and the end by saying "...physics schemes in WRF v3.8.1". This makes one think WRF v3.8.1 is the current version of WRF, which is obviously not the case (I believe there is a WRF v4.5 currently). I think this needs to be clarified. It may also be necessary to mention why WRF v.3.8.1 is used rather than the current version. And if there are any differences, the authors should briefly describe those as well.

Reply: Sorry for the misleading expression.
In fact the work of model improvement mainly took place during June to July, 2017. At that time the WRF v3.9 had been released before long and lacked sufficient testing. So we chose the WRF v3.8.1, which was a recent and reliable version at that time. We compared the current version WRFv4.5 with WRFv3.8.1. The main differences are the black carbon (BC) as well as organic carbon and BC biomass burning aerosol emissions were added to Thompson microphysics scheme since WRF v4.4 (https://www2.mmm.ucar.edu/wrf/users/physics/mp28_updated_new.html). Another difference is the default GHGs data used in WRF has changed from RCP4.5 to SSP2-4.5
(https://www2.mmm.ucar.edu/wrf/users/wrf_users_guide/build/html/physics.html).
Besides, there are no other differences relating to the external forcing settings between the current WRFv4.5.1 and the WRFv3.8.1. Some explanations for the model version used in this study were added in section 2.1 Model improvements.

12) - L122-124: What is the criteria for considering water-friendly aerosols and ice-friendly aerosols as anthropogenic and natural aerosols respectively? If there any studies that has shown this, please cite them.
Reply: Thank you for the suggestion. The water-friendly aerosols and ice-friendly aerosols are divided based on the criteria of the Thompson microphysics schemes (Thompson and Eidhammer, 2014). In this physical scheme, water-friendly aerosols mainly consist of sulfate aerosols, sea salt, and organics aerosols, and ice-friendly aerosols are mainly dust aerosols. Usually, dust aerosols and sea salt are treated as natural aerosols, sulfate aerosols are mainly emitted by human industrial activities, and organic aerosols have either natural sources or anthropogenic sources (e.g., Li et al., 2016). In East Asia, the concentration of sulfate aerosols is much larger than the

organic aerosols and sea salt, so the water-friendly aerosols can be treated as anthropogenic aerosols. The relating references have been cited in the original text.

13) - L133: Please give examples of studies that have used those two datasets for research and model evaluation.
Reply: Thank you for the suggestion. We added some references in the text (Belda et al., 2015; Guo et al., 2020; Hu et al., 2018; Wu et al., 2017; Wu et al., 2022; Zhu and Yang, 2020).

References:
Wu X , Hao Z , Zhang Y ,et al. 2022. Anthropogenic influence on compound dry and hot events in China based on Coupled Model Intercomparison Project Phase 6 models.International Journal of Climatology, 42(8), 4379-4390.

Belda M, Eva H, Tomas K, et al. 2015. Evaluation of CMIP5 present climate simulations using the Koppen-Trewartha climate classification.Climate research, 64(3), 201-212.

Hu Z, Zhou Q, Chen X,et al. 2018. Evaluation of three global gridded precipitation data sets in central Asia based on rain gauge observations. International Journal of Climatology, 38(9), 3475-3493.

Wu J, Gao X, Giorgi F, et al. 2017. Changes of effective temperature and cold/hot days in late decades over China based on a high resolution gridded observation dataset. International Journal of Climatology, 37, 788-800.

Zhu Y, Yang S. 2020. Evaluation of CMIP6 for historical temperature and precipitation over the Tibetan Plateau and its comparison with CMIP5. Advances in Climate Change Research, 11(3), 239-251.

Guo Z, Fang J, Sun X, et al. 2020. Decadal long convection-permitting regional climate simulations over eastern China: evaluation of diurnal cycle of precipitation. Climate Dynamics, 54(3), 1329-1349.

14) - L159-161: This is only true for the southeastern part of the domain. I think it should be mentioned.
Reply: Please see the following two figures. The left panel is the difference between climatological mean (1971-2000) annual mean temperature simulated by P1-CTRL and P2-CTRL, and the right panel is for the difference of annual precipitation. It is clear that P1 is warmer than P2 in the whole EC region, and P1 is also wetter than P2 in EC (also true for the northern part of EC: 35-45°N, 105-123°E, the regional mean annual precipitation difference in northern EC is 20.1 mm).

[Figure]

[Figure]

15) - L180: external
Reply: The word "extern" has been changed to "external".

16) - L184: weaken
Reply: The word "weakening" has been changed to "weaken".

17) - L185-187: This is very well known already. Different parameterization schemes can considerably alter simulation results.
Reply: Yes. We are inclined to leave this sentence to keep coherence.

18) - L189-193: Here the authors suggest that "spectral nudging can improve temperature trend simulation…". In my opinion, this is not robust since the authors did not investigate it for the other configurations.
Reply: Yes, this is not a general conclusion. We added "in our case" to the end of the sentence.

19) - L196-198: This will be clearer/robust if there is a P2-GHG and a P2-GHG-NNG as well.
Reply: Thank you for the comments. This is also because the lack of computing resources. We are trying to request more funding to extend this study.

20) - L208-209: I am not sure if this is entirely true. In my opinion, this is true when considering only the sign of the trend but not the magnitude. I think the forcing configurations can have a considerable localized influence on the magnitude of the trend. If you zoom into the southern areas of the box shown on the P1 figures, you can notice some changes in the magnitudes of the trend. This should be clarified.
Reply: Thank you for the suggestion. We revised this sentence to "These similar results implying the influence of forcing configurations inside the RCM on the precipitation trend was not obvious (mainly influence local trend magnitude but not large scale trend sign)".

21) - L232-234: A robust conclusion cannot be made based on just the 2 choices used here. I suggest the authors revise this so that it does not seem like this result is generalizable.

Reply: Thank you for the suggestion. We revised this sentence to "The P1-CTRL and P2-CTRL simulated the same increasing temperature trend (0.16 °C/10a). It implies may be the selection of physical parameterization schemes had no significant impact on the annual temperature trend in EC, but more parameterization schemes should be tested in the future".

22) - L246: change "So, the large…" to "So, a large…"

Reply: The word "the" has been changed to "a".

23) - L254: cautions to caution

Reply: The word "cautions" has been changed to "caution".

24) - L250: (63.4 mm/10a) Is this value reported in Table 2? Because I cannot find it. I see -20.7 for P2-CTRL. Please check and clarify.

Reply: The value 63.4mm/10a is the absolute difference between P2-CTRL and P2-NNG in table 2. To make it more clear, we added the above explanation after the value in brackets.

25) - L256-257: The authors should make it clear which data source is being spoken about here. Is it the observations? If yes, which one? They should also ensure that the values reported in the text correspond with those in the Table.

Reply: Sorry for the wrong value. The summer precipitation trend here is 9.3mm/10a, which is the trend of CN05.1. We revised this sentence to "The long-term trend of summer precipitation for CN05.1 in EC was 9.3 mm/10a, slightly larger than its annual precipitation trend (3.1 mm/10a) but was not significant, either.".

26) - L279: region to regions

Reply: The word "region" has been changed to "regions".

27) - L294: remove in

Reply: The word "in" has been removed.

28) - L330: Did the authors mean Clausius-Clapeyron?

Reply: Sorry for the wrong spelling. The name has been revised.

29) - L339: area to areas

Reply: The word "area" has been changed to "areas".

30) - L345: contributed to contribute

Reply: The word "contributed" has been changed to "contribute".

31) - L362-364: I think this sentence should be revised, and possibly broken down into two, as it is a bit difficult to understand.

Reply: Thank you for the suggestion. We have broke this sentence into two sentences "During our study period, there were five volcanic eruptions could be recognized in total. Among them the eruptions in 1963, 1982, and 1991 were stronger than the climatological mean level (Figure 7a)."

32) - L364: remove obviously

Reply: The word "obviously" has been removed.

33) - L370-371: The reason why there needs to be more ensemble members to evaluate internal variability. This should be addressed somehow in the text at least.

Reply: Thank you for the suggestion. We think the lateral boundary condition can suppress the model internal variability on the decadal time scale. Although the influence of model internal variability may be more significant on interannual time scale, this is beyond the scope of this paper, as we mainly focus on the influence of external forcing on long-term RCM simulation. We added some illustrations in the discussion section to explain why we did not perform the ensemble simulation.

34) - L377: change overestimate to overestimation

Reply: The word "overestimate" has been changed to "overestimation".

35) - L386: remove seems

Reply: The word "seems" has been removed.

36) - L396: change records to record

Reply: The word "records" has been changed to "record".

37) - L412: change resulted to result

Reply: The word "resulted" has been changed to "result".

**Referee 2:**

Comments to the Author

The authors modified the WRF v3.8.1 to include the spatial-temporal varying external forcing components to study the impact of dynamical forcing on the long-term simulation in China. Though the objective of the work is fascinating, I have some serious concerns about the WRF modifications and the experimental design.

1. One of the major points of this work is the development of WRF to include the external forcing components in WRF. But authors spent just one paragraph (Model improvement) under the 'Model and Data' section, which is also very general. They did not spend a single sentence about their improvements compared to the current

version of the WRF (WRF v4.5.1). Why did the authors modify the WRF v3.8.1 is also unclear and explained in the manuscript? WRF v3.8.1 was released in Aug 2016, so why we need to improve that 7-year-old version compared to the current version is also not explained.

Reply: Thank you for the comments.

In fact the work of model improvement mainly took place during June to July, 2017. At that time the WRF v3.9 had been released before long and lacked sufficient testing. So we chose the WRF v3.8.1, which was a recent and reliable version at that time. We compared the current version WRFv4.5 with WRFv3.8.1. The main differences are the black carbon (BC) as well as organic carbon and BC biomass burning aerosol emissions were added to Thompson microphysics scheme since WRF v4.4 (https://www2.mmm.ucar.edu/wrf/users/physics/mp28_updated_new.html). Another difference is the default GHGs data used in WRF has changed from RCP4.5 to SSP2-4.5

(https://www2.mmm.ucar.edu/wrf/users/wrf_users_guide/build/html/physics.html).

Besides, there are no other differences relating to the external forcing settings between the current WRFv4.5.1 and the WRFv3.8.1. Some explanations for the model version used in this study were added in section 2.1 Model improvements.

2. My second primary concern of this manuscript is its experimental design. The authors recommended judging the need for the nudging technique before the formal simulation, but their experimental design for nudging is surprising to me. Under the P1 subset, authors performed a control simulation, followed by all forcings and individual forcings of GHGs, Aerosol, and Volcano. All the experiments mentioned above were performed using Nudging techniques, but suddenly then, they performed only a single simulation without Nudging experiments (for GHGs). What is the scientific basis for choosing GHGs forcing without Nudging experiments? Why not choose all forcing experiments for Nudging justification? I recommend that authors perform without Nudging experiments, at least for all forcings and Aerosol forcings on top of GHG forcings. Also, for the P2 experiment, they only performed a control simulation and nudging experiment, but their major objective was to see the impact of external forcings.

Reply: Thank you for the constructive comments. We designed our experiments mainly for the following reasons:

1) The main purpose of this paper is to explore the potential impact of adding dynamical external forcing in the WRF model. And previous studies showed that spectral nudging can significantly improve the simulation results in East Asia (e.g., ). Therefore, we consider using spectral nudging is a common choice for most RCM users. We did some test for different parameterization schemes and nudging options mainly for providing a reference to the impact of dynamical external forcing. We also know that performing nudging test on all different scenarios, and comparing more physical parameterization schemes are better, but it will significantly increase the computation cost.

2) We did not set a simulation of P1-NFC-NNG (using P1 physical schemes, no forcing in the RCM, no spectral nudging), because this is a common choice for most RCMs users (it means most researchers do not use dynamical external forcing, maybe some of them use spectral nudging, and their choices for physical schemes are not limited for the choice of this paper).

3) We compared the effect of spectral nudging for P1-GHG, because GHGs forcing is added to WRF model since version 3.5, the WRF users can turn on this option without the modification of WRF model. Therefore, the combination of spectral nudging and GHGs forcing is a potential choice for all the WRF users, and currently such evaluations is insufficient, so we did this thing. Another reason is GHGs are the most importance external forcing, and other forcing may have contradict effect to GHGs such as aerosols. What we want to know is the potential of the maximum difference between the simulations of using nudging and not using nudging, and focusing on the most important forcing is our best choice.

4) We also compared the influence of spectral nudging for P2-CTRL, because this is the situation of considering all the forcing components. In theory, using all kinds of forcing is better than using a single forcing. Currently there are rarely works have discussed the usage of nudging will change the simulation result to what extend when using all kinds of forcing in RCM. Although the simulations here are far from sufficient enough, we think the comparison between P2-CTRL and P2-NNG is also beneficial to give a preliminary answer to this question. We did not expect to test the influence of nudging on all the situations (and this is also unrealistic), as the main purpose of this paper is to evaluate the impact of adding dynamical external forcing in the RCM.

5) To better explain the experiment design, we reorganized the section 2.3 experiment design. Please see the updated manuscript.

3. Panel 'b' in Figures 3 – 6 shows the combined effects of GHG and Aerosol on the temperature, precipitation, and circulation trend. I want another experiment by adding GHG and Aerosol as external forcings. I believe that impact might have differed from panel 'b' in Figures 3 – 6 since they are not in a linear relationship in the actual atmosphere. We might not speculate some conclusions by linearly adding to independent single forcing simulations specifically for precipitation.

Reply: Thank you for the comments. We know that in reality the relationship between different forcing responses are not linear. In fact, it depends on the spatial scales. At large scales (such as global scale or continental scale), the forcing responses are usually considered linearly addible. This is also one of the basic assumptions of the detection and attribution methods such as optimal fingerprint method. According to Figures 3-6, we can see that in our research region, the impact of GHGs and aerosols are generally linearly additive. Some local differences in precipitation trends are inevitable between the linearly added results and the two-forcing simulation results, because precipitation usually has large spatial variabilities than other meteorological variables.

My other points are also listed below.

4. This point is related to my first major concern. How is your modification different from the most recent WRF version (WRF 4.5.1) regarding aerosol contribution through Thompson Microphysics and RRTMG radiation scheme?

Reply: Same as the reply to the first comment.

The main differences between WRFv3.8.1 and WRFv4.5.1 are the black carbon (BC) as well as organic carbon and BC biomass burning aerosol emissions were added to Thompson microphysics scheme since WRF v4.4 (https://www2.mmm.ucar.edu/wrf/users/physics/mp28_updated_new.html). Another difference is the default GHGs data used in WRF has changed from RCP4.5 to SSP2-4.5 (https://www2.mmm.ucar.edu/wrf/users/wrf_users_guide/build/html/physics.html). Besides, there are no other differences relating to the external forcing settings between these two versions.

5. Authors must provide more detail about their modification and sensitivity with and without modification experiments, at least for a single sensitivity.

Reply: Thank you for the suggestion. We added more descriptions about our modifications in the manuscript:

The modifications involves different levels of the WRF software framework. We first modified the Registry.EM_COMM file to define the newly added or modified variables and physical packages, as well as the new namelist options. Then we modified the top level (relating files: solve_em.F, module_first_rk_step_part1.F, module_configure.F) and the driver level (relating files: module_surface_driver.F, module_microphysics_driver.F, module_radiaGon_driver.F) in WRF model in order to make sure our modifications in physical schemes could be called correctly. Finally, we modified the physical level of the model to ingest time varying external forcing data or add new physical processes (relating files: module_mp_thompson.F, module_ra_rrtmg_sw.F, module_ra_rrtmg_lw.F, module_ra_cam.F).

These descriptions are added to the second paragraph of section 2.1. A power point of model improvement details and some sensitivity tests are also uploaded as a supplementary information file 2.

6. Why are authors using the ERA40 reanalysis, not the most recent ERA5, to force the WRF model? ERA5 provides much better spatial and temporal resolution compared to ERA40.

Reply: We use the ERA40 reanalysis because this work started in 2017, and at that time the ERA5 had not been released yet.

7. Similar to the above point, why authors are using CMIP5 forcings instead of CMIP6 data? These can make a significant difference in their simulations.

Reply: Also because this is an old work, the CMIP6 had not been released at that time.

8. Figure S4 is very much ambiguous, and it's difficult to quantify which scheme performs better other than by eye estimation. Authors should do a more quantitative way to choose the best scheme, and authors can try the Taylor Diagram for U, Geopotential, Temperature, and Precipitation.

Reply: Thank you for the comments. We did some quantitative evaluations before the selection of parameterization schemes. For example, the figure below shows the Talyor Diagram of the summer precipitation rate (mm/day) of 1998 in EC region (105-123°E, 20-45°N) for different tests and TRMM (reference). As we can see, the two parameterization scheme groups used in our experiments (i.e., MYJ-BMJ-ng, YSU-SAS-ng) are generally better than other groups (MYJ-ZM-ng is located in the second quadrant, with negative correlation coefficient). Other groups also have relative good performance, such as MYNN3-BMJ-ng and YSU-BMJ-ng. However, these groups have larger bias outside the EC region, particular in southwestern and northeastern China (Figure S4). Therefore, considering the results of Taylor Diagram and the large-scale rain-belt pattern, we chose the group of YSU-SAS-ng and MYJ-BMJ-ng.

[Figure]

9. To quantify the Nudging effect robustly, authors must perform without nudging experiments for all forcings (P-MFC-NNG) and aerosol forcing (P1-AERO-NNG). This will help them to quantify the nudging impact.

Reply: Thank you for the suggestions. Considering these experiments were conducted several years ago, and our computing platform has been changed during these years, any little differences may lead to different simulation results even for repeating the same experiment. Therefore we think adding new simulations may not be beneficial.

10. For precipitation trend analysis, how reliable are CRU and ERA40 precipitation data?

Reply: According to a research in 2006, the reliability of CRU is higher than ERA40 for summer precipitation in China (Zhao and Fu, 2006). Considering summer precipitation account for most proportion of the annual precipitation, this conclusion may also true for the annual precipitation.

Zhao T, Fu C. 2006. Comparison of products from era40, ncep-2, and cru with station data for summer precipitation over china. Advances in Atmospheric Sciences, 23(4), 593-604.